# Myosin light chain 3 serves as a receptor for nervous necrosis virus entry into host cells via the macropinocytosis pathway

Lan Yao[1,2,3†], Wanwan Zhang[1,2,3†], Xiaogang Yang[1,2,3], Meisheng Yi[1,2,3]*, Kuntong Jia[1,2,3]*

[1]School of Marine Sciences, Sun Yat-sen University, Zhuhai, China; [2]Southern Marine Science and Engineering Guangdong Laboratory, Zhuhai, China; [3]Guangdong Provincial Key Laboratory of Marine Resources and Coastal Engineering, Guangzhou, China

**\*For correspondence:**
yimsh@mail.sysu.edu.cn (MY);
jiakt3@mail.sysu.edu.cn (KJ)

[†]These authors contributed equally to this work

**Competing interest:** The authors declare that no competing interests exist.

## eLife Assessment

The findings in this manuscript are **fundamental** because they identify an entry receptor MYL3 that belongs to the myosin family as a possible target that could inhibit a virus that has a high impact on aquaculture. The evidence is **convincing** as it contains strong in vitro and in vivo data that support their conclusions; however, studies on the presence of MYL3 in NNV target tissues will further strengthen their claims

**Abstract** Nodaviridae infections cause severe mortality in insects and fish, with nervous necrosis virus (NNV) posing significant threats to global fish populations. However, the host factors involved in NNV entry remain poorly understood. We identify myosin light chain 3 from marine medaka (*Oryzias melastigma*) (MmMYL3) as a novel receptor for red-spotted grouper NNV (RGNNV), facilitating internalization via macropinocytosis. MmMYL3 directly binds the RGNNV capsid protein (CP), which depends on the arm and S domains of CP and the EF-hand2 domain of MmMYL3. In vitro experiments showed that MmMYL3 siRNA, protein, anti-MYL3 antibodies, or the arm domain synthetic peptides blocked RGNNV entry. Moreover, in vivo administration of MmMYL3 protein also inhibited RGNNV infection. Ectopic MmMYL3 expression enabled RGNNV internalization into resistant cells. Notably, MmMYL3 facilitated RGNNV internalization through the macropinocytosis pathway via the IGF1R-Rac1/Cdc42 axis. Collectively, our findings underscore MYL3's crucial role in NNV entry and its potential as an antiviral target.

## Introduction

Viral encephalopathy and retinopathy, also known as viral nervous necrosis, is a highly contagious and pathogenic disease that infects more than 120 marine and freshwater fish species, leading to mass mortalities and severe economic losses in the aquaculture industry. The etiological agent responsible, nervous necrosis virus (NNV), belongs to the Nodaviridae family, genus *Betanodavirus* and manifests as a nonenveloped single-stranded RNA virus (*Costa and Thompson, 2016*; *Munday et al., 2002*). The NNV genome is composed of two RNA segments: RNA1 encodes the RNA-dependent RNA polymerase (RDRP), while RNA2 encodes the viral capsid protein (CP), which plays a critical role in the initial interaction with host cell surface receptors, thus facilitating viral entry (*Bandín and Souto, 2020*; *Mori et al., 1992*).

Viral entry into host cells is a key determinant of host specificity, tissue tropism, and viral pathogenicity, and involves intricate interactions between viral CPs and host cell receptors (*Grove and Marsh, 2011*). Viruses utilize sophisticated strategies to attach to one or multiple receptors, enabling them to cross the plasma membrane and access the necessary host cell machinery (*Maginnis, 2018*). Viral receptors function not only as attachment moieties but also as entry factors, coordinators of viral trafficking, and activators of signaling events (*Marsh and Helenius, 2006*). Therefore, identifying virus receptors and understanding the mechanism of virus-receptor interaction is critical for better preventing viral diseases. Although NNV exhibits a broad host tropism, only heat shock cognate protein 70 (HSC70), heat shock protein 90ab1 (HSP90ab1), and Nectin1 have been identified as its receptor or co-receptor (*Chang and Chi, 2015*; *Zhang et al., 2020*; *Zhang et al., 2024*). Following NNV CP recognition of the cellular surface receptor, the NNV enters the cell by receptor-mediated endocytosis. Viruses usually take advantage of various endocytic pathways to enter host cells, with clathrin-mediated endocytosis (CME), macropinocytosis, and caveolar/raft-dependent endocytosis being the most extensively studied (*Mercer et al., 2010*). Previous studies revealed that NNV particles utilize the receptor-mediated CME and macropinocytosis internalization pathways to invade host cells (*Liu et al., 2005*), with the former facilitated by HSP90ab1 and HSC70 (*Zhang et al., 2020*). However, the receptor responsible for mediating NNV invasion of host cells through macropinocytosis remains unknown.

Among the diverse cellular factors that can serve as viral receptors, members of the myosin family have emerged as intriguing candidates (*He et al., 2022*). The myosin family comprises a wide range of actin-associated motor proteins that play a role in a variety of cellular activities, including signal transduction, intracellular transport, cell division, etc. Up to date, more than 40 known classes of myosin have been identified (*Foth et al., 2006*). Recent studies underscore their significance in various viral infections. For instance, myosin heavy chain 9 has been identified as a crucial factor for various virus infections (*Li et al., 2018*). Non-muscle myosin heavy chain IIA serves as a functional entry receptor for herpes simplex virus-1 (*Arii et al., 2010*). Additionally, myosin II light chain activation is essential for influenza A virus cell entry through macropinocytosis (*Banerjee et al., 2014*).

Myosin light chain 3 (MYL3), a constituent of the myosin light chain family predominantly found in muscle tissues, particularly the heart and skeletal muscles, plays a pivotal role in muscle contraction by forming the myosin complex with myosin heavy chain and facilitating actin filament sliding (*Ingles et al., 2005*; *Zhang et al., 2016*). Mutations in the MYL3 gene have been associated with various muscle-related disorders, including cardiomyopathies and skeletal muscle myopathies (*Mavilakandy and Ahamed, 2022*). However, the role of MYL3 in virus infection remains undiscovered. Huang et al. reported that *Epinephelus coioides* MYL3 might interact with native NNV CP by proteomic analysis of immunoprecipitation (IP) assay (*Huang et al., 2020*). In our previous study, we also found that marine medaka (*Oryzias melastigma*) MYL3 (MmMYL3) is a potential interacting protein of the purified CP (*Zhang et al., 2020*), yet its involvement in NNV infection, particularly its role in viral invasion, remains uncertain.

In this study, we unveil MmMYL3 as a novel NNV entry receptor, elucidating its facilitation of RGNNV entry into host cells through macropinocytosis, mediated by the IGF1R-Rac1/Cdc42 pathway. These findings offer novel insights into the intricate mechanisms underlying NNV entry and highlight potential targets for the development of antiviral drugs.

## Results

### Interaction between MmMYL3 and CP

First, a Co-IP assay was conducted to validate the interaction between MmMYL3 and CP. MmMYL3 exhibited co-localization and co-precipitation with CP (*Figure 1A and B*). GST-tag pull-down assays provided additional confirmation of the direct interaction between MmMYL3 and CP, as well as MmMYL3 and RGNNV virions (*Figure 1C and D*). Surface plasmon resonance (SPR) analysis further confirmed a direct interaction between MmMYL3 and CP, with kinetics revealing association rate constant (Ka), dissociation rate constant (Kd), and equilibrium dissociation constant ($K_D$) of $1.448 \times 10^4$ $M^{-1}$ $s^{-1}$, $1.679 \times 10^{-4}$ $s^{-1}$, and 2.95 μM, respectively (*Figure 1E*). Additionally, the interaction between CP and MYL3 was consistently observed in another marine fish species, *Lateolabrax japonicus* (*Figure 1F*). Furthermore, our data indicated that MmMYL3 did not interact with the CP of

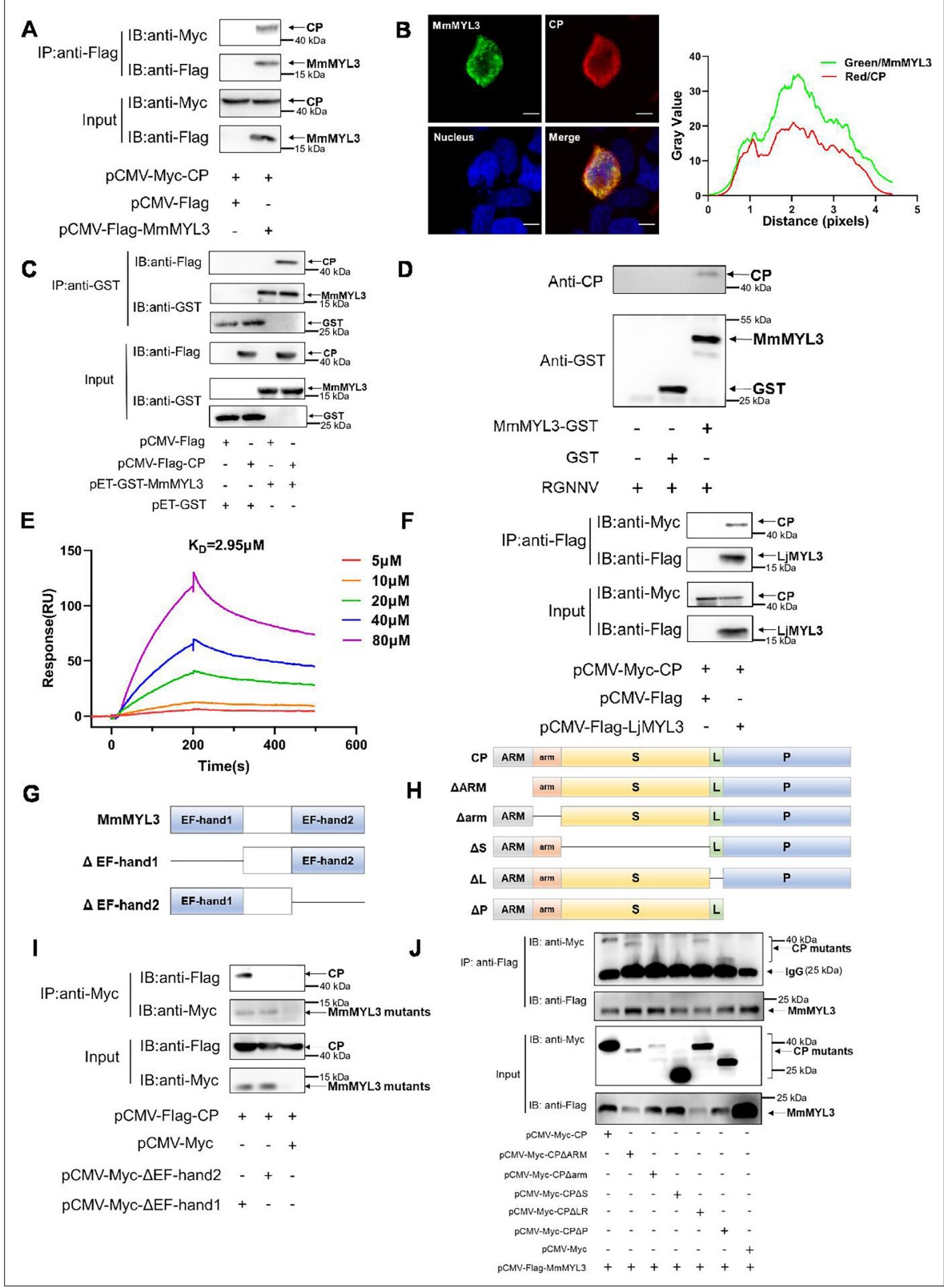

**Figure 1.** MmMYL3 interacts with CP. (**A**) Immunoprecipitation (IP) (with anti-Flag) and immunoblot analysis (with anti-Flag and anti-Myc) of HEK293 cells transfected with plasmids encoding MmMYL3-Flag and CP-Myc for 48 hr. (**B**) HEK293T cells were co-transfected with MmMYL3-Flag and CP-Myc plasmids. MmMYL3 (green) and CP (red) were detected by immunofluorescence staining with anti-Flag or anti-Myc abs, respectively. Nucleus was stained by DAPI, bar = 10 µm. (**C**) The lysates of HEK293T cells transfected with indicated plasmids were pulled down with purified MmMYL3-GST

*Figure 1 continued on next page*

*Figure 1 continued*

or GST proteins followed by immunoblotting with anti-GST and anti-Flag abs. (**D**) Pull-down of RGNNV virions by MmMYL3-GST or GST proteins. RGNNV, purified proteins, and magnetic streptavidin beads were co-incubated as indicated. Pull-down was performed with a magnet, and the pellet was subjected to western blot analysis. (**E**) Surface plasmon resonance (SPR) analysis of MmMYL3-CP interaction. A 1:1 binding model was used to calculate the $K_D$. (**F**) Plasmids carrying MYL3 from sea perch (LjMYL3) were transfected into HEK293T cells, together with CP-Myc plasmid. At 48 hr post-transfection, the cell lysates were subjected to Co-IP analysis with anti-Flag magnetic beads as indicated. (**G**) Schematic diagram of MmMYL3 and its truncated mutants. (**H**) Schematic diagram of CP and its truncated mutants. (**I**) IP (with anti-Myc) and immunoblot analysis (with anti-Flag and anti-Myc) of HEK293 cells transfected with plasmids encoding CP-Flag and Myc-tagged full-length MmMYL3 or its truncated mutants for 48 hr. (**J**) HEK293T cells were co-transfected with MmMYL3-Flag and different Myc-tagged CP mutants for 48 hr. Co-IP assays were performed as described above.

The online version of this article includes the following source data and figure supplement(s) for figure 1:

**Source data 1.** Original files for western blot analysis displayed in *Figure 1A*.

**Source data 2.** PDF file containing original western blots for *Figure 1A*, indicating the relevant bands and treatments.

**Source data 3.** Original data of *Figure 1B*.

**Source data 4.** Original files for western blot analysis displayed in *Figure 1C*.

**Source data 5.** PDF file containing original western blots for *Figure 1C*, indicating the relevant bands and treatments.

**Source data 6.** Original files for western blot analysis displayed in *Figure 1D*.

**Source data 7.** PDF file containing original western blots for *Figure 1D*, indicating the relevant bands and treatments.

**Source data 8.** Original data of *Figure 1E*.

**Source data 9.** Original files for western blot analysis displayed in *Figure 1F*.

**Source data 10.** PDF file containing original western blots for *Figure 1F*, indicating the relevant bands and treatments.

**Source data 11.** Original files for western blot analysis displayed in *Figure 1I*.

**Source data 12.** PDF file containing original western blots for *Figure 1I*, indicating the relevant bands and treatments.

**Source data 13.** Original files for western blot analysis displayed in *Figure 1J*.

**Source data 14.** PDF file containing original western blots for *Figure 1J*, indicating the relevant bands and treatments.

**Figure supplement 1.** MmMYL3 could not interact with CP of covert mortality nodavirus (CMNV).

**Figure supplement 1—source data 1.** Original files for western blot analysis displayed in *Figure 1—figure supplement 1*.

**Figure supplement 1—source data 2.** PDF file containing original western blots for *Figure 1—figure supplement 1*, indicating the relevant bands and treatments.

covert mortality nodavirus (CMNV) (*Figure 1—figure supplement 1*). These results robustly affirm the binding affinity of MmMYL3 to CP.

## Domain mapping of the association between MmMYL3 and CP

To delineate the domain of MmMYL3 required for interaction with CP, various Myc-tagged distinct domains of MmMYL3 were constructed, and Co-IP assays revealed that deletion of the EF-hand2 domain of MmMYL3 abolished its interaction with CP, suggesting the essential role of the EF-hand2 domain (*Figure 1G and I*). Furthermore, to ascertain which domain of CP is necessary for interaction with MmMYL3, we generated a series of Myc-tagged CP deletion mutants (*Figure 1H*). Co-IP assays revealed that deletion of the arm domain and S domain of CP resulted in the abrogation of its interaction with MmMYL3 (*Figure 1J*). Collectively, these findings indicate that MmMYL3 interacts with the arm and S domain of CP through its EF-hand2 domain.

## Localization of MmMYL3 on the cell surface

We next explored whether MmMYL3 is present on the cell surface, a prerequisite for a functional viral receptor. Immunofluorescence (IF) assays in permeabilized and non-permeabilized hMMES1 and HEK293T cells transfected with Myc-MmMYL3 revealed MmMYL3 localization both at the cell surface and in the cytoplasm (*Figure 2A and B*). Meanwhile, no immunofluorescent signal was detected on the surface of non-permeabilized cells using anti-Actin abs. Conversely, in permeabilized cells treated with Triton X-100, MmMYL3 proteins were found not only on the cell surface but in the cytoplasm. To corroborate the cell surface localization of MmMYL3, a protease protection assay was carried out. Besides, the addition of proteinase K resulted in the significant reduction of MmMYL3 protein in non-permeabilized HEK293T or hMMES1 cells transfected with pCMV-Myc-MmMYL3 plasmid compared

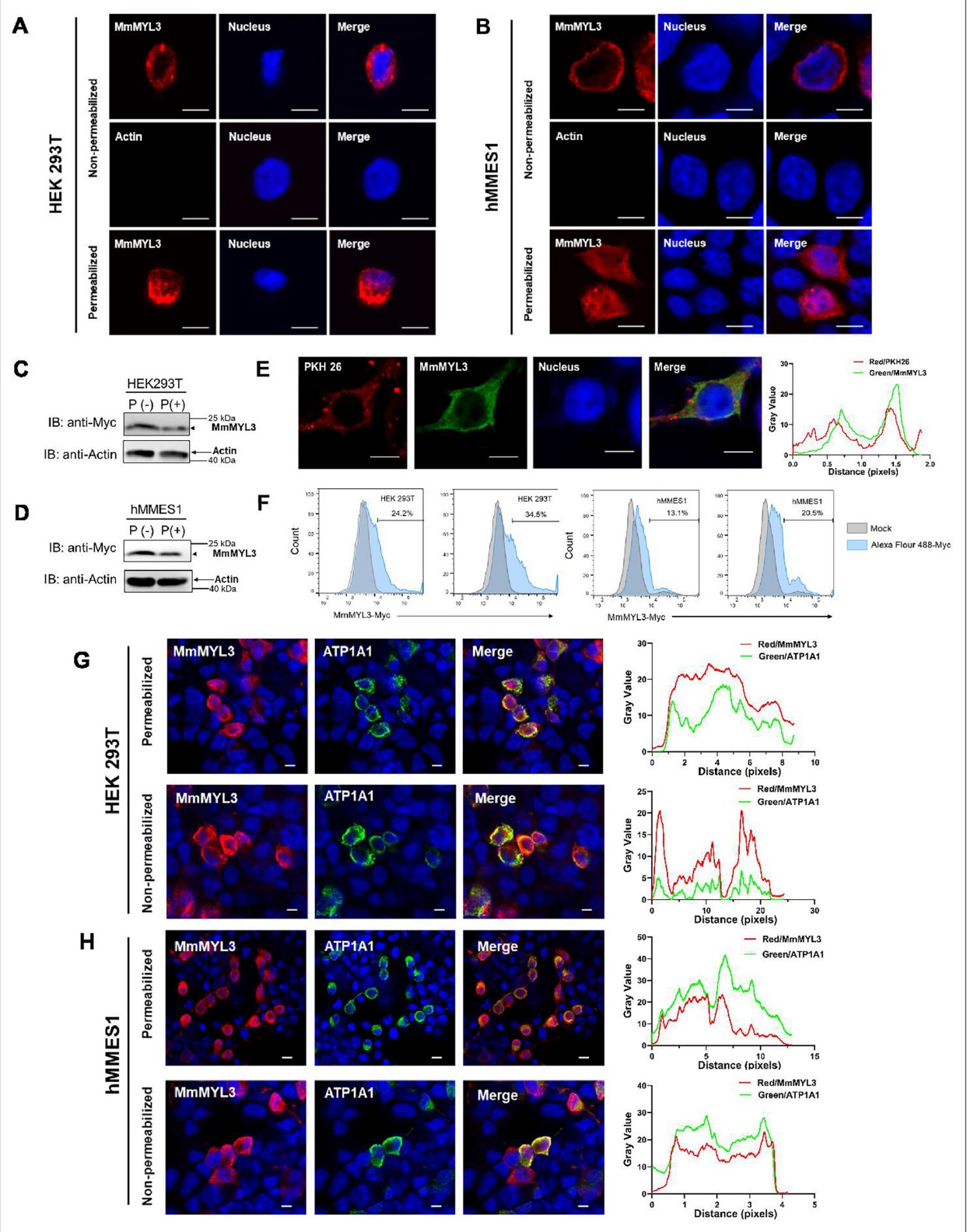

**Figure 2.** Surface localization of MmMYL3 in HEK293T and hMMES1 cells. (**A and B**) HEK293T (**A**) or hMMES1 (**B**) cells transfected with MmMYL3-Myc plasmids were fixed with formalin, treated with Triton X-100 or not, and immunostained with anti-Myc abs or anti-Actin abs, respectively. Cell nuclei were stained with DAPI. Bar = 10 μm. (**C and D**) HEK293T and hMMES1 cells transfected with MmMYL3-Myc plasmid were treated with proteinase K (P+) or without proteinase K (P-) and harvested for a western blot analysis using anti-Myc and anti-Actin abs. (**E**) hMMES1 cells transfected with MmMYL3-Myc

*Figure 2 continued on next page*

*Figure 2 continued*

plasmid were labeled with PKH 26 (red) and stained with anti-Myc abs (green). Nucleus was stained by DAPI, bar = 10 μm. (**F**) HEK293T and hMMES1 cells transfected with Myc-MmMYL3 were stained with Alexa Fluor 488-conjugated anti-Myc abs. Surface expression of MmMYL3 was analyzed by flow cytometry. (**G and H**) HEK293T or hMMES1 cells were co-transfected with MmMYL3-Myc and ATP1A1-Flag plasmids. MmMYL3 (red) and ATP1A1 (green) were detected by immunofluorescence staining with anti-Myc or anti-Flag abs in permeabilized or non-permeabilized conditions. Nucleus was stained by DAPI, bar = 10 μm.

The online version of this article includes the following source data for figure 2:

**Source data 1.** Original files for western blot analysis displayed in *Figure 2C and D*.

**Source data 2.** PDF file containing original western blots for *Figure 2C and D*, indicating the relevant bands and treatments.

**Source data 3.** Original data of *Figure 2E*.

**Source data 4.** Original data of *Figure 2F*.

**Source data 5.** Original data of *Figure 2G*.

**Source data 6.** Original data of *Figure 2H*.

to the control without proteinase K treatment (*Figure 2C and D*), confirming that MmMYL3 is exposed on the surface of these cells.

To visualize the co-localization of MmMYL3 with the cell membrane, PKH 26, a red fluorescent dye that binds stably to the lipid region of the cell membrane, was used. Co-localization of MmMYL3 with PKH 26 in hMMES1 cells was observed, further corroborating its membrane association (*Figure 2E*). Flow cytometry analysis using Alexa Fluor 488-conjugated anti-Myc antibody also indicated a proportional increase in MmMYL3 surface expression with increasing MmMYL3 concentrations in HEK293T and hMMES1 cells (*Figure 2F*). Lastly, co-transfection of MmMYL3 with the membrane protein Na$^+$/K$^+$-ATPase alpha 1 (ATP1A1) in HEK293T and hMMES1 cells showed clear co-localization of MmMYL3 with ATP1A1 in both permeabilized and non-permeabilized conditions (*Figure 2G and H*), further affirming the presence of MmMYL3 on the cell surface. Collectively, these findings robustly demonstrate that MmMYL3 is localized on the cell surface.

## MmMYL3 serves as a surface receptor of RGNNV

To investigate whether MmMYL3 was involved in RGNNV infection, we first examined the expression pattern of *MmMYL3* during RGNNV infection. *MmMYL3* exhibited a significant high expression from 2 to 48 hr post-infection (hpi), peaking at 4 hpi, indicating MmMYL3 might play a vital role in the early stages of RGNNV infection (*Figure 3—figure supplement 1A*). Ectopic expression of MmMYL3 significantly increased *CP* expression and RGNNV titer (*Figure 3A–C*), while MmMYL3 knockdown resulted in decreased *CP* gene transcription and virus titer (*Figure 3D–F*).

To further substantiate that MmMYL3 functions as a receptor for RGNNV infection, commercial antihuman MYL3 abs and purified GST-tagged MmMYL3 proteins were used to evaluate the role of MmMYL3 in the virus entry process. Notably, recombinant MmMYL3 (*Figure 3—figure supplement 1B*) proteins significantly reduced the entry of RGNNV into hMMES1 cells in a dose-dependent manner (*Figure 3G and H*), and treatment with anti-MYL3 abs (*Figure 3—figure supplement 1C*) resulted in a significant reduction in *CP* and *RDRP* levels at 2 and 4 hpi, suggesting inhibition of RGNNV entry (*Figure 3I and J*). Moreover, synthetic peptides derived from the arm domain of CP could block RGNNV entry in a dose-dependent manner (*Figure 3K and L*).

Cellular receptors required for virus infection confer infectability when they are expressed in an uninfectable cell type. To further demonstrate the receptor function of MmMYL3, we tested whether overexpression of MmMYL3 could render typically resistant cells susceptible to RGNNV infection. HEK293T cells, which are not naturally permissive to RGNNV, were transfected with MmMYL3 or MmHSP90ab1 (a known NNV receptor). First, we detected CP anti-reverse sequences (*CP* (-)), which is a replicative intermediate for the production of viral RNAs, in MmMYL3 or MmHSP90ab1-overexpressing HEK293T cells post RGNNV infection, but not in empty vector-transfected cells (*Figure 4A*). Similar to MmHSP90ab1, overexpression of MmMYL3 promoted the entry of RGNNV into HEK293T cells (*Figure 4B*). The internalization of RGNNV into MmMYL3-overexpressing HEK293T cells was further evidenced by the intracellular IF localization of CP at 24 hpi (*Figure 4C*). Additionally, like observations in hMMES1 cells, abundant viral particles were observed in the cytoplasm of MmMYL3-overexpressing

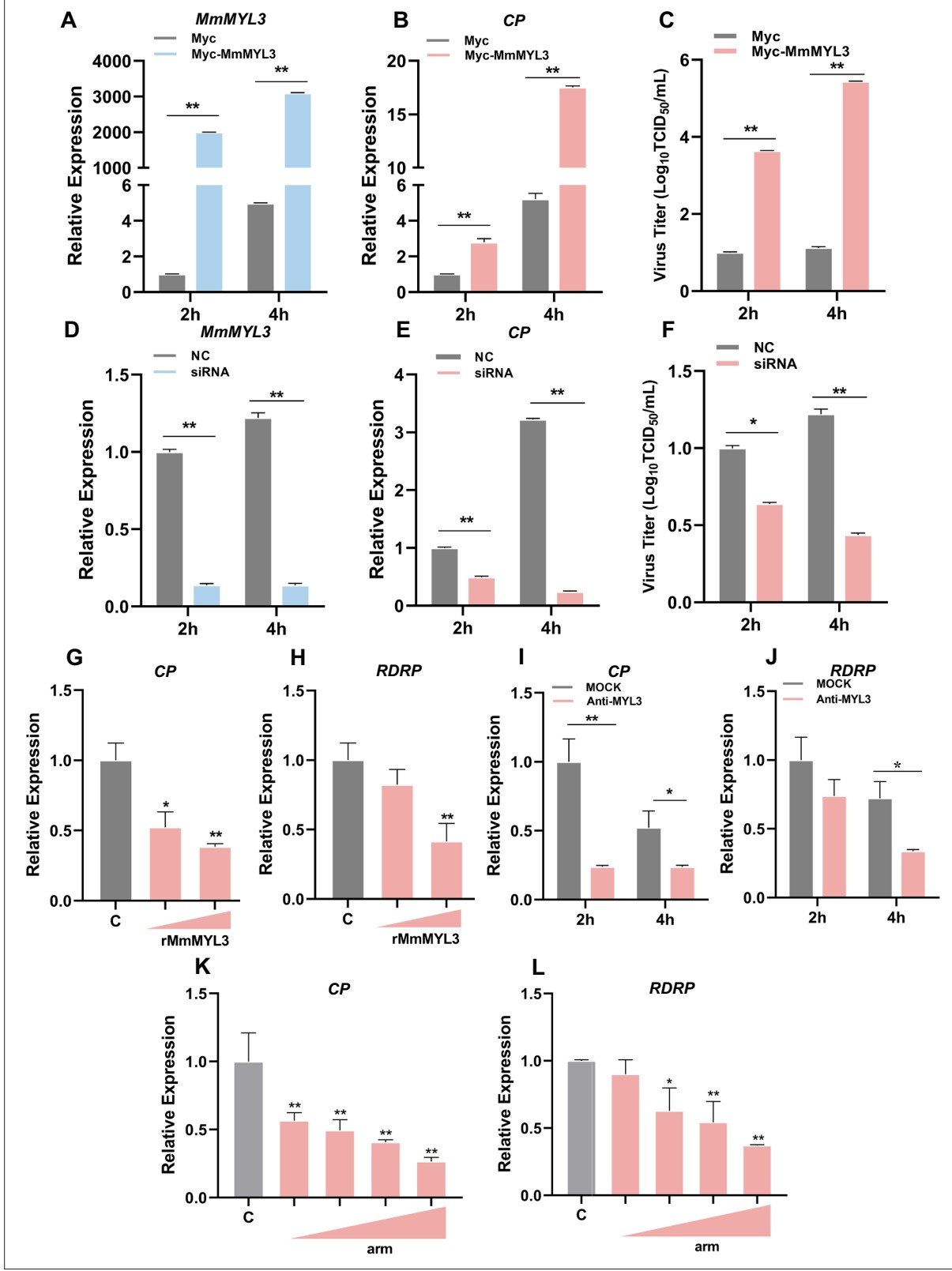

**Figure 3.** Effect of MmMYL3 on RGNNV entry. (**A and B**) hMMES1 cells were transfected with MmMYL3-Myc or Myc plasmid (control) and infected with RGNNV (multiplicity of infection [MOI] = 1) for 2 and 4 hr, respectively. Then, the cells were lysed for quantitative reverse transcription PCR (qRT-PCR) to detect the expression of MmMYL3 and CP. (**C**) Virus titer detection in MmMYL3-Myc-transfected hMMES1 cells post RGNNV infection for 2 and 4 hr. (**D and E**) qRT-PCR analysis of MmMYL3 and CP expression in siMmMYL3- or control-transfected hMMES1 cells infected with RGNNV for 2

*Figure 3 continued on next page*

*Figure 3 continued*

and 4 hr. (**F**) Virus titer detection in siMmMYL3-transfected hMMES1 cells post RGNNV infection for 2 and 4 hr. (**G and H**) RGNNV was incubated with purified MmMYL3-GST proteins (100 or 500 ng) for 4 hr at 4°C, then was added to hMMES1 cells which were further incubated for 4 hr at 4°C. Cells were washed with PBS three times and harvested for CP (**G**) and RNA-dependent RNA polymerase (RDRP) (**H**) expression detection. (**I and J**) hMMES1 cells were incubated with commercial anti-human MYL3 abs (1:50) for 4 hr and then infected with RGNNV for 2 or 4 hr at 4°C. After being washed with PBS three times, cells were harvested for CP (**I**) and RDRP (**J**) expression detection. (**K and L**) hMMES1 cells were treated with different concentrations of arm peptides for 2 hr at 4°C, and after RGNNV infection for 4 hr, CP (**K**) and RDRP (**L**) expression were detected by qRT-PCR. The results are presented as mean ± SD. Statistical significance was determined by an unpaired two-tailed Student's t test. *p<0.05, **p<0.01. Data are representative of three independent experiments.

The online version of this article includes the following source data and figure supplement(s) for figure 3:

**Source data 1.** Original data of *Figure 3*.

**Figure supplement 1.** The role of MmMYL3 in RGNNV infection.

**Figure supplement 1—source data 1.** Original files for western blot analysis displayed in *Figure 3—figure supplement 1B*.

**Figure supplement 1—source data 2.** PDF file containing original western blots for *Figure 3—figure supplement 1B*, indicating the relevant bands and treatments.

**Figure supplement 1—source data 3.** Original files for western blot analysis displayed in *Figure 3—figure supplement 1C*.

**Figure supplement 1—source data 4.** PDF file containing original western blots for *Figure 3—figure supplement 1C*, indicating the relevant bands and treatments.

HEK293T cells post RGNNV infection (*Figure 4D*). These results further support the susceptibility conferred by MmMYL3 expression, highlighting its role as a receptor for RGNNV.

## MmMYL3 is vital for RGNNV pathogenesis in marine medaka

To evaluate the physiological relevance of MmMYL3 in RGNNV infection, marine medaka were injected with mixtures of RGNNV and recombinant MmMYL3 protein or GST control protein, respectively (*Figure 4E*). Fish injected with the MmMYL3-RGNNV mixture exhibited significantly higher survival rates compared to those injected with the GST-RGNNV mixture (*Figure 4F*), indicating a protective effect of MmMYL3 on viral pathogenesis. Consistent with these findings, quantitative reverse transcription PCR (qRT-PCR) analysis showed that CP expression levels in the brain and eye tissues were markedly reduced in the medaka treated with the MmMYL3 protein (*Figure 4G*). Furthermore, histopathological analysis revealed reduced pathological injury in the eyes and brains of the fish treated with MmMYL3 protein, as evidenced by diminished vacuolization compared to the control group (*Figure 4H and I*). These results strongly indicate that MmMYL3 plays a critical role in mediating RGNNV infection and pathogenesis in marine medaka.

## Involvement of MmMYL3 in RGNNV entry via macropinocytosis

Previous studies have reported macropinocytosis and CME as the primary routes of NNV internalization. To determine whether macropinocytosis is involved in RGNNV entry into hMMES1 cells, we used a panel of compounds to treat the cells prior to viral infection, including EIPA (macropinocytosis inhibitor), Rottlerin (macropinocytosis inhibitor), CPZ (CME inhibitor), and Nystatin (caveolin inhibitor). After treatment, hMMES1 cells were infected with RGNNV, and viral copies in cell lysates were quantified by qRT-PCR. The results indicated that EIPA, Rottlerin, and CPZ significantly inhibited RGNNV infection compared to the dimethyl sulfoxide (DMSO) control (*Figure 5A*), whereas Nystatin had no obvious effect, indicating RGNNV enters hMMES1 cells via macropinocytosis and CME pathways.

To clarify the role of MmMYL3 in the process of RGNNV endocytosis, MmMYL3-overexpressing hMMES1 cells were treated with the previously mentioned inhibitors before virus infection. qRT-PCR experiments demonstrated a significant, dose-dependent reduction in internalized RGNNV with EIPA and Rottlerin treatments (*Figure 5B*). Fluid-phase uptake is a distinctive feature activated during macropinocytosis. Studies on various viruses entering host cells via macropinocytosis have demonstrated a transient enhancement in fluid-phase uptake (*Delpeut et al., 2017*). Dextran served as a macropinocytosis marker, being taken up by cells during macropinocytosis. As expected, cells infected with RGNNV displayed dextran-positive vesicles resembling larger, irregular macropinosomes at 4 hpi, in contrast to mock-infected cells (*Figure 5C*). Meanwhile, clusters of MmMYL3 were

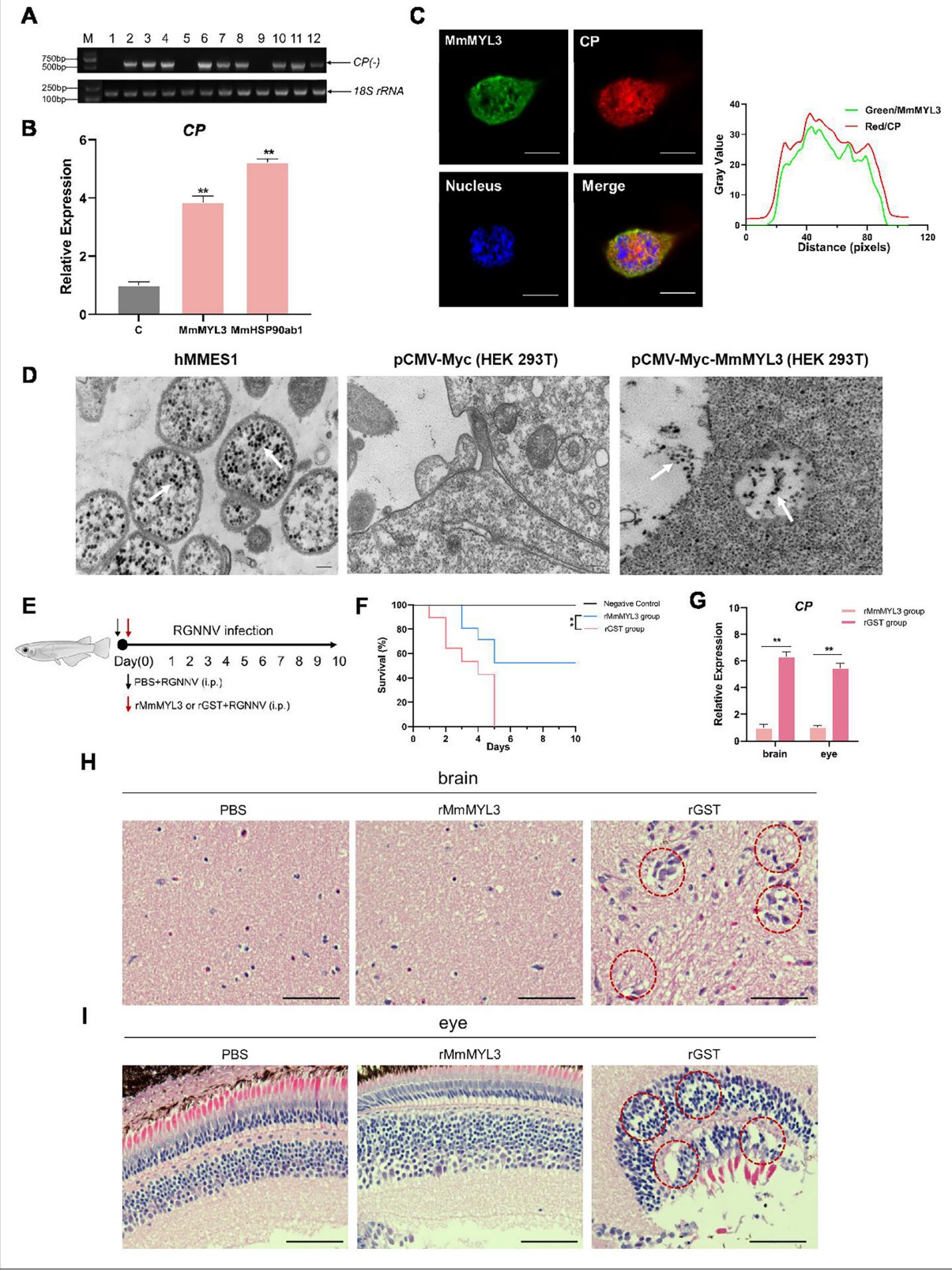

**Figure 4.** MmMYL3 was involved in RGNNV internalization, and recombinant MmMYL3 protein treatment protects marine medaka from RGNNV pathogenesis. (**A**) Reverse transcription PCR (RT-PCR) analysis of CP (-) sequence. HEK293T cells were transfected with Myc empty vector (lines 1, 5, and 9), MmMYL3-Myc (lines 2, 6, and 10), MmHSP90ab1-Myc (lines 3, 7, and 11), or both MmMYL3-Myc and MmHSP90ab1-Myc (lines 4, 8, and 12) plasmids for 24 hr, respectively. Then, cells were infected with RGNNV (multiplicity of infection [MOI] = 5) for 4 hr. Next, the cells were washed to remove any

*Figure 4 continued on next page*

*Figure 4 continued*

unbound viruses and incubated for 24 hr (lines 1–4), 48 hr (lines 5–8), and 72 hr (lines 9–12). Cells were harvested, and total RNA was extracted for CP (-) detection by RT-PCR. Human 18S rRNA was detected as a reference. (**B**) HEK293T cells were transfected with MmMYL3-Myc, MmHSP90ab1-Myc, or Myc plasmids, respectively. Then, transfected cells were infected with RGNNV (multiplicity of infection [MOI] = 5) for 24 hr. Next, the cells were washed to remove any unbound viruses, and total RNA was extracted for CP detection by quantitative RT-PCR (qRT-PCR). (**C**) HEK293T cells transfected with Myc empty vector or MmMYL3-Myc were infected with RGNNV (MOI = 5) for 24 hr. CP (red) and MmMYL3 (green) were detected by immunofluorescence staining. Cell nuclei were stained with DAPI. Bar = 10 µm. (**D**) Transmission electron micrograph of RGNNV-infected hMMES1 cells and HEK293T cells transfected with Myc empty vector or MmMYL3-Myc with 80,000 magnifications. Bar = 200 nm. (**E**) Schematic representation of the experimental design. RGNNV (100 TCID$_{50}$) was incubated with purified GST-tagged MmMYL3 recombinant protein (rMmMYL3, 500 ng) or GST protein (rGST, 500 ng) for 4 hr at 4°C. Marine medaka were infected with rMmMYL3-RGNNV or GST-RGNNV mixture by intraperitoneal (i.p.) injection, respectively. The negative control group of fishes was injected with the same volume of PBS. (**F**) Survival rates of marine medaka infected with RGNNV and MmMYL3 or GST protein mixtures. The cumulative survival rate was determined from 1 to 10 days post-infection. (**G**) Quantification of RGNNV CP expression in brain and eye tissues by qRT-PCR. (**H and I**) Histopathological analysis of the brain (**H**) and eye (**I**) tissues of fish treated with PBS, rMmMYL3, or rGST. Marine medaka were necropsied, and the eyes and brains were collected at day 5 post-infection. Hematoxylin and eosin (H&E) staining was used to assess tissue integrity. The vacuolization in eyes and brains was marked by a red dotted circle. Bar = 100 µm. Data were collected from three independent experiments and presented as mean ± SD. All results are representative of three similar experiments. **, p<0.01.

The online version of this article includes the following source data for figure 4:

**Source data 1.** Original files for western blot analysis displayed in *Figure 4A*.

**Source data 2.** PDF file containing original western blots for *Figure 4A*, indicating the relevant bands and treatments.

**Source data 3.** Original data of *Figure 4B*.

**Source data 4.** Original data of *Figure 4C*.

**Source data 5.** Original data of *Figure 4F*.

**Source data 6.** Original data of *Figure 4G*.

observed co-localized with RGNNV CP in dextran-positive macropinosomes (*Figure 5D*). These findings suggest that MmMYL3 facilitated RGNNV entry into hMMES1 cells via macropinocytosis.

## Involvement of MmIGF1R in MmMYL3-mediated RGNNV entry

Macropinocytosis initiation necessitates the coordinated activation of multiple signaling pathways, typically induced by growth factors that prompt the activation of receptor tyrosine kinases (RTKs) (*Zhang et al., 2022a*). To determine whether RTK activation is essential for RGNNV entry mediated by MmMYL3, a series of inhibitors of RTKs were utilized. Inhibitors of IGF1R (AZD-3463, Picropodophyllin) could block MmMYL3-mediated RGNNV entry, but inhibitors of InsR (NVP-TAE), EGFR (Gefitinib), and c-Met (SU11274) had no significant effect on virus entry mediated by MmMYL3 (*Figure 6A*). To further clarify the relationship between MmMYL3 and MmIGF1R, MmMYL3-Myc and MmIGF1R-GFP were co-transfected into HEK293T cells, and we found that MmMYL3 was co-localized with MmIGF1R (*Figure 6B*). Additionally, the Co-IP assay showed that MmIGF1R could interact with MmMYL3 through the extracellular domain (*Figure 6C and D*). Furthermore, the pull-down assay further validated the direct interaction between MmMYL3 and MmIGF1R (*Figure 6E*). Additionally, overexpression of MmIGF1R promoted the entry of RGNNV, whereas knockdown of MmIGF1R by small interfering RNA (siRNA) inhibited RGNNV entry (*Figure 6F–I*). These results strongly indicate that MmIGF1R is tightly associated with MmMYL3-mediated viral invasion.

## Rac1/Cdc42 contributes to RGNNV entry

The Ras superfamily GTPases, particularly Rac1 and Cdc42, are essential regulators of macropinocytosis, contributing to actin cytoskeletal reorganization and membrane ruffling (*Bar-Sagi and Feramisco, 1986*; *Bar-Sagi et al., 1987*). We therefore examined the relationship between Rac1, Cdc42, and IGF1R. Co-IP assays revealed the co-precipitation of MmRac1 and MmCdc42 with MmIGF1R (*Figure 7A and B*). Furthermore, we found that MmRac1 and MmCdc42 could interact with the intracellular region of MmIGF1R (*Figure 7C and D*). In addition, MmRac1 and MmCdc42 were observed to co-localize with MmIGF1R (*Figure 7E and F*). To evaluate the functional significance of MmRac1 and MmCdc42 in RGNNV internalization, hMMES1 cells were transfected with either MmRac1 or MmCdc42. Overexpression of either MmRac1 or MmCdc42 enhanced RGNNV internalization, as demonstrated by increased viral RNA levels (*Figure 7G–J*). In contrast, the specific inhibitors of Rac1

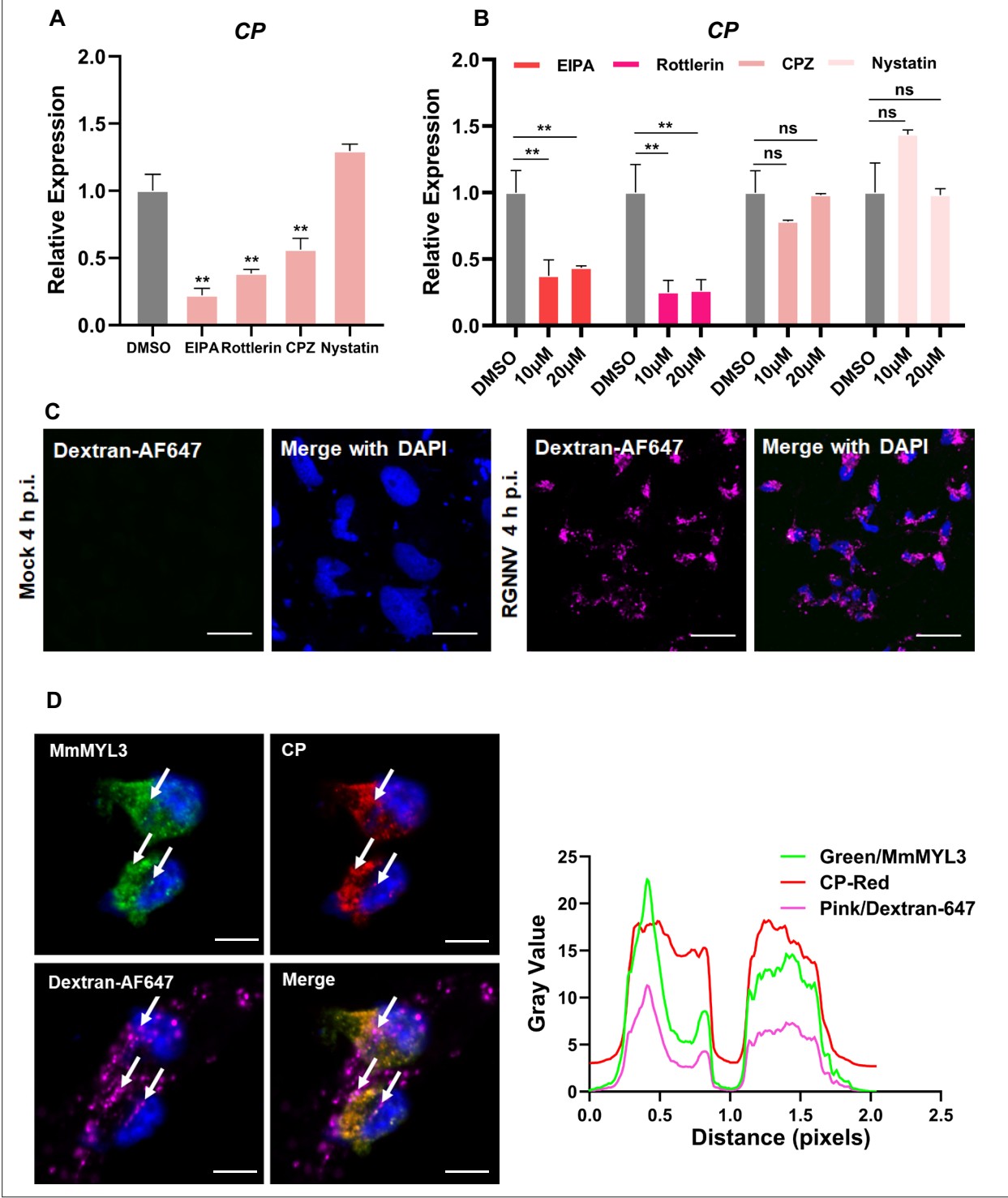

**Figure 5.** RGNNV triggers macropinocytosis mediated by MmMYL3. (**A**) Cells pretreated with EIPA, Rottlerin, CPZ, or Nystatin were infected with RGNNV (multiplicity of infection [MOI] = 1) for 4 hr at 28°C, and the expression of CP was analyzed by quantitative reverse transcription PCR (qRT-PCR). (**B**) MmMYL3-overexpressing hMMES1 cells were treated with different inhibitors and incubated with RGNNV (MOI = 1) for 4 hr at 28°C, and the expression of CP was analyzed by qRT-PCR. (**C**) hMMES1 cells were mock-infected or infected with RGNNV (MOI = 1) in medium containing Alexa Fluor 647-conjugated dextran (10,000 MW). At 4 hr post-infection (hpi), cells were fixed and nuclei counterstained with DAPI and imaged on a confocal microscope. Bar = 10 μm. (**D**) Co-localization of MmMYL3 (green), RGNNV CP (red), and dextran-AF647 in RGNNV-infected hMMES1 cells. Cell nuclei were stained with DAPI. Bar = 10 μm. Data were collected from three independent experiments and presented as mean ± SD. All results are representative of three similar experiments. ns, not significant; **, p<0.01.

*Figure 5 continued on next page*

*Figure 5 continued*

The online version of this article includes the following source data for figure 5:

**Source data 1.** Original data of *Figure 5A*.

**Source data 2.** Original data of *Figure 5B*.

**Source data 3.** Original data of *Figure 5D*.

(EHT1864) and Cdc42 (MLS-573151) significantly reduced RGNNV entry in a dose-dependent manner (*Figure 7K and L*).

Actin filaments play a crucial role in facilitating membrane folding and vesicle formation during macropinocytosis, thereby influencing the efficiency and regulation of this cellular uptake mechanism. Thus, we further monitored actin rearrangement during RGNNV infection in hMMES1 cells. hMMES1 cells were stained with DAPI and iFluor-488-conjugated phalloidin at 4 hpi, specifically binding to the polymerized form of actin (F-actin). Intriguingly, RGNNV infection led to depolymerization and changes in actin distribution. The number of F-actin filaments decreased, and actin-driven membrane protrusions co-localized with virions were observed on the cell surface of infected cells, absent in mock-infected control cells (*Figure 7M*). These results suggest that MmRac1 and MmCdc42 facilitate RGNNV internalization by driving macropinocytosis through actin cytoskeletal dynamics.

## Discussion

NNV, an important fish virus, enters host cells by utilizing a range of host factors that act as receptors and facilitate the process of endocytosis via an orchestrated mechanism. Three host factors, HSP90ab1, HSC70, and Nectin1, have been identified as entry receptors of NNV (*Chang and Chi, 2015*; *Zhang et al., 2020*, n.d.); however, disrupting their interaction with cells does only partially hinder viral infection, suggesting that NNV likely utilizes additional, yet unidentified host receptors or co-receptors. To establish infection, NNV utilized receptor-mediated endocytosis to induce internalization of the virus/receptor complex, primarily including CME and macropinocytosis (*Liu et al., 2005*). Although previous studies demonstrate that HSP90ab1 and HSC70 are responsible for NNV internalization via CME, the receptor and the molecular mechanisms leading to NNV entry into cells via macropinocytosis remain to be determined. In this study, we found that NNV gains entry into host cells via the interaction between CP and its novel receptor MmMYL3. Subsequently, MmMYL3 engages with IGF1R, leading to the activation of the Rac1/Cdc42 pathway, which induces actin rearrangement and facilitates NNV macropinocytosis (*Figure 7N*).

Numerous data indicate the significance of myosin light chains in the functionality of both cardiac and skeletal muscles. However, little is known about the function of myosin light chains during virus infection (*Kamm and Stull, 2011*; *Sheikh et al., 2016*; *Yu et al., 2016*). Here, we provide several lines of evidence showing that MmMYL3 acts as a novel receptor of NNV. First, ectopic expression of MmMYL3 facilitates internalization of RGNNV, which is mediated by the interaction between the EF-hand2 domain of MmMYL3 and the arm and S domain of CP. Considering that our previous study had reported that MmHSP90ab1, a known NNV receptor, interacted with the linker region of CP (*Zhang et al., 2020*). Thus, we speculated that NNV might have the ability to utilize multiple receptors simultaneously for cell infection through these receptors binding to the different domains of CP. Furthermore, interference with the MmMYL3 and CP interaction with abs against MYL3, MmMYL3 fusion protein, and CP arm domain peptides significantly blocks NNV entry, providing a potential antiviral strategy against NNV infection. In addition to these in vitro findings, our in vivo experiments further confirm the physiological relevance of MmMYL3 in viral pathogenesis and highlight its potential as a therapeutic target in aquaculture. More importantly, virus-resistant HEK293T cells are rendered susceptible to RGNNV infection by overexpression of MmMYL3 alone. The development of MmMYL3-overexpressing HEK293T cell model offers a valuable platform for investigating viral infection mechanisms and conducting drug screening studies. All these results suggested that MmMYL3 is a novel receptor that mediates NNV entry. We also found that the interaction of MYL3 with CP was conserved across NNV-sensitive fish. Interestingly, NNV exhibits a broad species tropism, capable of infecting diverse cell lines and various animal species examined thus far (*Munday et al., 2002*). This phenomenon may be attributed in part to the robust conservation of MYL3. To the best

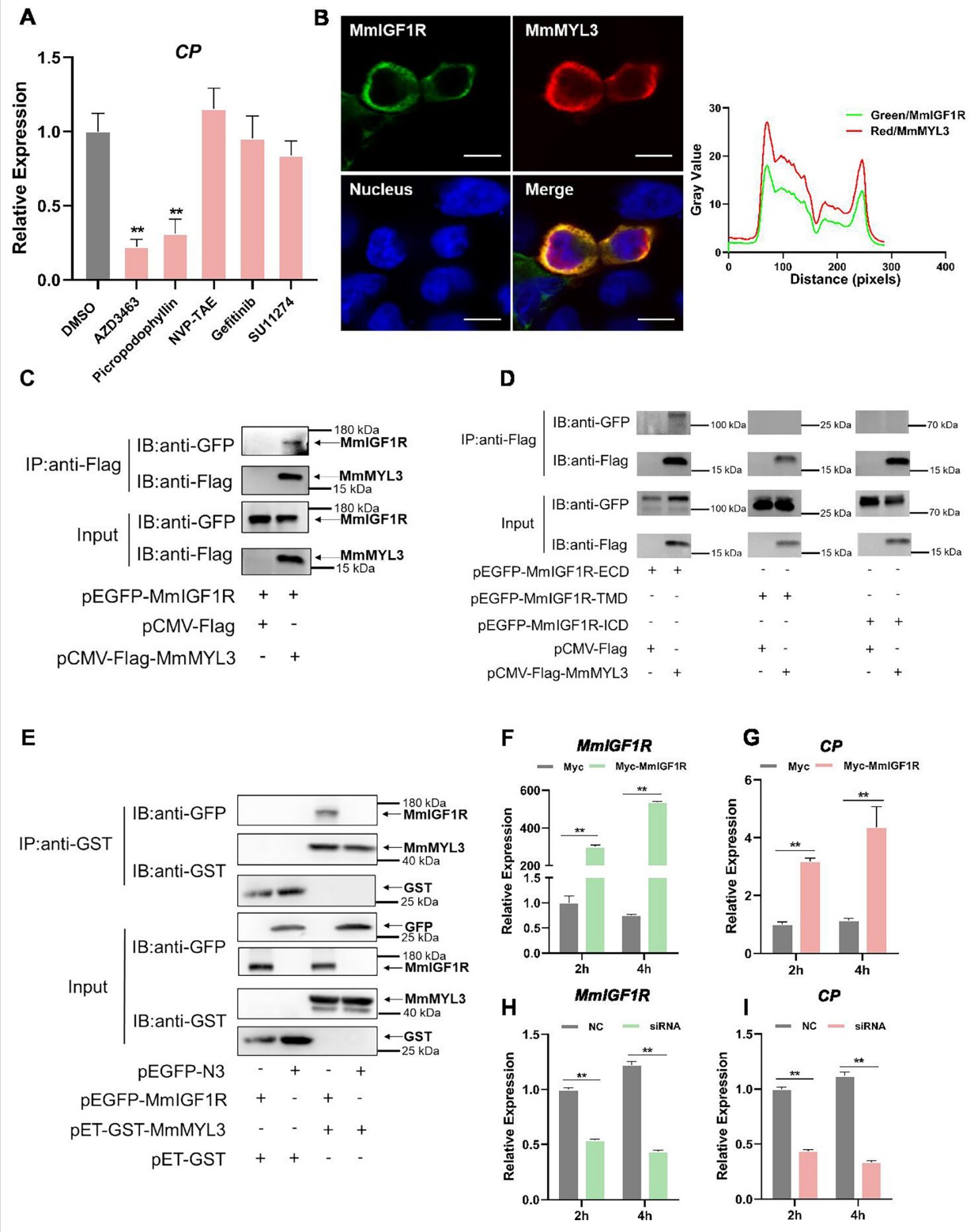

**Figure 6.** MmIGF1R is involved in MmMYL3-mediated RGNNV internalization. (**A**) MmMYL3-overexpressing hMMES1 cells were treated with different receptor tyrosine kinases (RTKs) inhibitors and incubated with RGNNV for 4 hr at 28°C, and the expression of CP was analyzed by quantitative reverse transcription PCR (qRT-PCR). (**B**) MmIGF1R-GFP and MmMYL3-Myc were transfected into HEK293T cells as indicated for immunofluorescence analysis by using anti-Myc (red) abs. Nuclei were stained with DAPI. Bar = 10 μm. (**C and D**) Immunoprecipitation (IP) (with anti-Flag) and immunoblot analysis (with anti-Flag and anti-GFP) of HEK293 cells transfected with plasmids encoding MmMYL3-Flag, MmIGF1R-GFP, MmIGF1R-ECD-GFP, MmIGF1R-TMD-GFP,

*Figure 6 continued on next page*

*Figure 6 continued*

or MmIGF1R-ICD-GFP for 48 hr. (**E**) The lysates of HEK293T cells transfected with indicated plasmids were pulled down with purified MmMYL3-GST or GST proteins. The proteins bound to MmIGF1R, and the inputs were immunoblotted with anti-GST and anti-GFP abs. (**F and G**) hMMES1 cells were transfected with MmIGF1R-GFP or GFP-N3 plasmid (control) and infected with RGNNV (multiplicity of infection [MOI] = 1) for 2 and 4 hr, respectively. Then, the cells were lysed for qRT-PCR to detect the expression of MmIGF1R and CP. (**H and I**) qRT-PCR analysis of MmIGF1R and CP mRNA expression of siMmIGF1R- or NC (control)-transfected hMMES1 cells, following infection with RGNNV for 2 and 4 hr. Data were collected from three independent experiments and presented as mean ± SD. All results are representative of three similar experiments. **, p<0.01.

The online version of this article includes the following source data for figure 6:

**Source data 1.** Original data of *Figure 6A*.

**Source data 2.** Original data of *Figure 6B*.

**Source data 3.** Original files for western blot analysis displayed in *Figure 6C*.

**Source data 4.** PDF file containing original western blots for *Figure 6C*, indicating the relevant bands and treatments.

**Source data 5.** Original files for western blot analysis displayed in *Figure 6D*.

**Source data 6.** PDF file containing original western blots for *Figure 6D*, indicating the relevant bands and treatments.

**Source data 7.** Original files for western blot analysis displayed in *Figure 6E*.

**Source data 8.** PDF file containing original western blots for *Figure 6E*, indicating the relevant bands and treatments.

**Source data 9.** Original data of *Figure 6F*.

**Source data 10.** Original data of *Figure 6G*.

**Source data 11.** Original data of *Figure 6H*.

**Source data 12.** Original data of *Figure 6I*.

of our knowledge, this is the first study to provide evidence that MYL3 functions as a functional entry receptor for virus.

Nodaviridae viruses are divided into two genera: *Alphanodavirus* and *Betanodavirus*. *Alphanodaviruses* typically infect insects, whereas *Betanodaviruses* primarily infect fish (*Yong et al., 2017*). In recent years, a novel *Alphanodavirus*, provisionally named CMNV, has emerged, causing viral covert mortality disease in shrimp and also affecting farmed marine fish (*Wang et al., 2021*; *Zhang et al., 2014*). Given the susceptibility of marine fish to CMNV, we investigated the possibility that both *Alphanodavirus* and *Betanodavirus* utilize MYL3 as a common receptor for cellular entry. Sequence comparison results revealed a low homology between the CP sequences of CMNV and RGNNV. Additionally, our data indicated that the CP of CMNV did not interact with MmMYL3. These findings suggest that MmMYL3 may function as a receptor specific to *Betanodavirus*, rather than being a common receptor for both viral genera. Receptor recognition is typically highly specific, and the binding interactions between viral proteins and host receptors often depend on the structural compatibility between the viral capsid/viral envelope and the host receptor. Since MYL3 does not interact with CMNV, a virus more closely related to NNV, it is less likely to function as a receptor for viruses that are more distantly related to NNV.

Previous reports have highlighted the dependence of the NNV entry process on both CME and macropinocytosis pathways (*Liu et al., 2005*). We had demonstrated that NNV can use MmHSP90ab1 and MmHSC70 to enter cells via the CME pathway (*Chang and Chi, 2015*; *Zhang et al., 2020*). In contrast to these receptors, our findings revealed that MmMYL3 promotes NNV internalization via macropinocytosis. Macropinocytosis, characterized as a transient, growth factor-induced, actin-dependent endocytic pathway, facilitates the internalization of fluid and membrane into large vacuoles (*Mercer and Helenius, 2009*). Mounting evidence underscores its pivotal role as an endocytic route exploited by numerous viruses for invading host cells (*Huang et al., 2008*; *Locker et al., 2000*; *Mercer and Helenius, 2008*). Upon binding to the cell membrane, viruses trigger the activation of RTKs or other signaling molecules such as Rac1, Cdc42, and Pak1, initiating intricate intracellular signaling cascades that lead to alterations in actin filament dynamics and induction of plasma membrane ruffling (*Bar-Sagi and Feramisco, 1986*; *Bar-Sagi et al., 1987*; *Letizia Lanzetti APLAS and PPDF, 2004*). RTKs, well-known upstream regulatory factors of macropinocytosis, are involved in the entry process of various viruses (*Amyere et al., 2000*; *Nicola et al., 2005*). To evaluate the significance of RTKs in NNV entry, we conducted a screening of small molecules targeting the activation of specific RTKs. Our findings revealed that AZD-3463 and Picropodophyllin, two inhibitors of

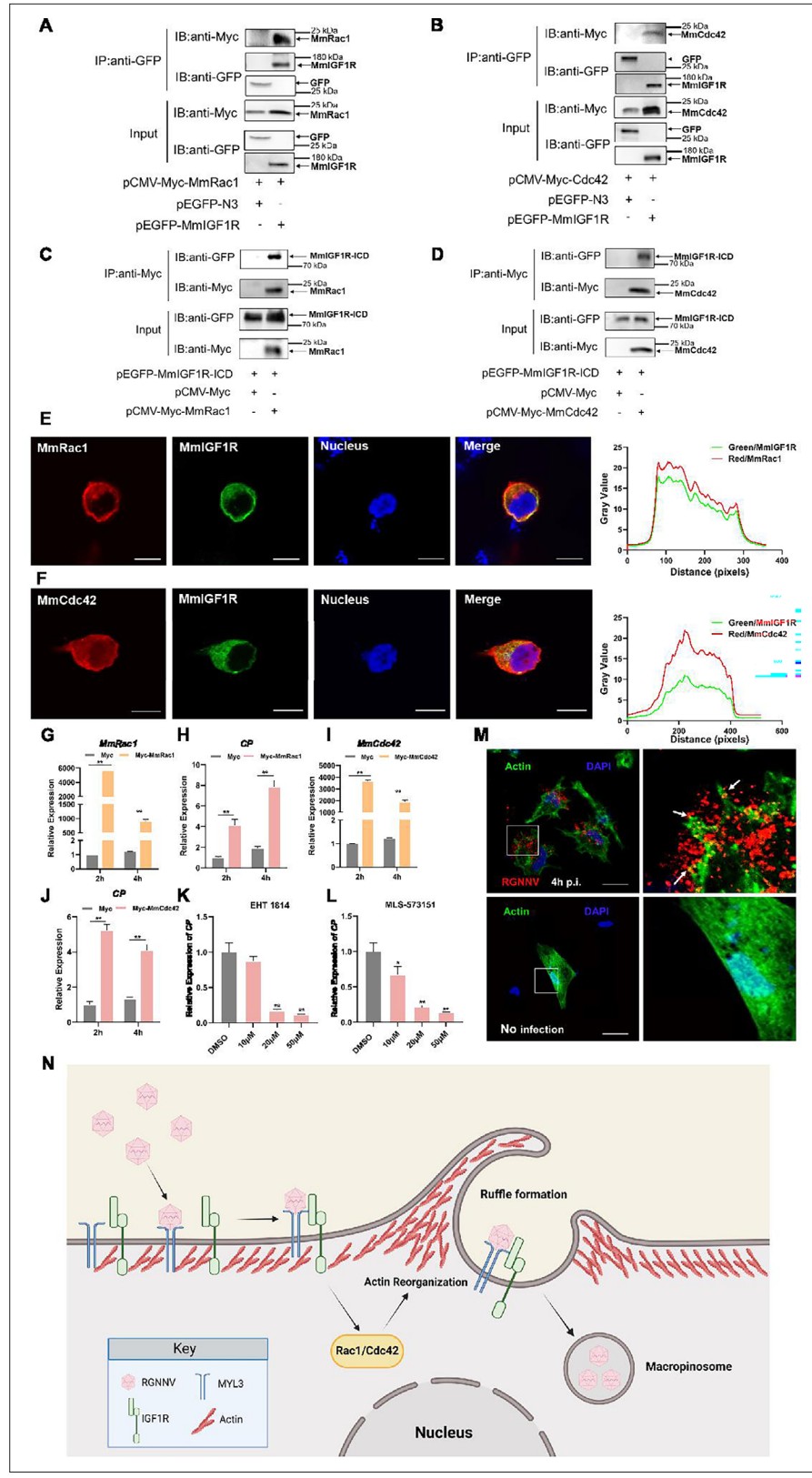

**Figure 7.** MmRac1 and MmCdc42 regulate RGNNV entry. (**A and B**) HEK293T cells were transfected with MmIGF1R-GFP and MmRac1-Myc or MmCdc42-Myc plasmids as indicated for 48 hr, and the cell lysates were subjected to coimmunoprecipitation analysis with anti-GFP magnetic beads as described above. (**C and D**) HEK293T cells were transfected with MmRac1-Myc, MmCdc42-Myc, and MmIGF1R-ICD-GFP plasmids,

*Figure 7 continued on next page*

*Figure 7 continued*

respectively. At 48 hr post-transfection, the cell lysates were subjected to coimmunoprecipitation analysis with anti-Myc magnetic beads as indicated. (**E and F**) MmIGF1R-GFP and MmRac1-Myc or MmCdc42-Myc were transfected into HEK293T cells as indicated for immunofluorescence analysis by using anti-Myc (red) antibodies. Nuclei were stained with DAPI. Bar = 10 μm. (**G–J**) hMMES1 cells were transfected with MmRac1-Myc, MmCdc42-Myc, or pCMV-Myc (control) plasmids and infected with RGNNV (multiplicity of infection [MOI] = 1) for 2 and 4 hr, respectively. Then, the cells were lysed for quantitative reverse transcription PCR (qRT-PCR) to detect the expression of MmRac1, MmCdc42, and CP. (**K and L**) hMMES1 cells were treated with inhibitors of Rac1 (EHT1814) or Cdc42 (MLS-573151) for 4 hr, then infected with RGNNV (MOI = 1) for 4 hr at 28°C, and the cells were lysed for qRT-PCR to detect the expression of CP. (**M**) Actin cytoskeleton dynamics are involved in RGNNV entry. hMMES1 cells were exposed to RGNNV (MOI = 1) or PBS for 4 hr, and actin filaments were labeled with iFluor 488 phalloidin (green), then fixed and permeabilized as described above. Nuclei were stained with DAPI. Images were captured with a ×100 oil immersion objective. A higher magnification of the boxed area reveals the formation of actin protrusions at the cell surface membrane (arrows). Bar = 10 μm. Data were collected from three independent experiments and presented as mean ± SD. All results are representative of three similar experiments. *, p<0.05; **, p<0.01. (**N**) Model of RGNNV entry into host cells via macropinocytosis mediated by MmMYL3. RGNNV particles engage hMMES1 cells through interactions between CP and MYL3, a receptor on the cell surface, which is followed by interactions between MYL3 and IGF1R. Then, IGF1R activates and recruits small GTPases Rac1/Cdc42 to induce actin reorganization and facilitate the internalization of RGNNV via macropinocytosis.

The online version of this article includes the following source data for figure 7:

**Source data 1.** Original files for western blot analysis displayed in *Figure 7A*.

**Source data 2.** PDF file containing original western blots for *Figure 7A*, indicating the relevant bands and treatments.

**Source data 3.** Original files for western blot analysis displayed in *Figure 7B*.

**Source data 4.** PDF file containing original western blots for *Figure 7B*, indicating the relevant bands and treatments.

**Source data 5.** Original files for western blot analysis displayed in *Figure 7C*.

**Source data 6.** PDF file containing original western blots for *Figure 7C*, indicating the relevant bands and treatments.

**Source data 7.** Original files for western blot analysis displayed in *Figure 7D*.

**Source data 8.** PDF file containing original western blots for *Figure 7D*, indicating the relevant bands and treatments.

**Source data 9.** Original data of *Figure 7E*.

**Source data 10.** Original data of *Figure 7F*.

**Source data 11.** Original data of *Figure 7G*.

**Source data 12.** Original data of *Figure 7H*.

**Source data 13.** Original data of *Figure 7I*.

**Source data 14.** Original data of *Figure 7J*.

**Source data 15.** Original data of *Figure 7K*.

**Source data 16.** Original data of *Figure 7L*.

---

IGF1R, could impede MmMYL3-mediated RGNNV entry. IGF1R is a transmembrane RTK involved in mediating cellular responses to IGFs and plays a significant role in cellular processes such as growth, proliferation, and survival (*Hakuno and Takahashi, 2018*; *Pellegrino et al., 2024*). Recent studies have revealed an intriguing connection between IGF1R signaling and viral entry mechanisms, particularly through macropinocytosis (*Józefiak et al., 2021*; *Stewart et al., 2021*). Our findings revealed direct binding between MmMYL3 and MmIGF1R, indicating that MmMYL3 might participate in the macropinocytosis-mediated NNV entry through its interaction with MmIGF1R. Previous study had identified IGF1R as an entry receptor for respiratory syncytial virus (*Griffiths et al., 2020*). Consider the conservation of IGF1R, we investigated whether it could serve as a potential receptor for NNV. However, our analysis indicated that MmIGF1R did not directly bind to CP, indicating that MmIGF1R might not be the receptor for NNV.

To further explore how MmIGF1R promotes macropinocytosis and NNV internalization, we investigate the effect of MmIGF1R downstream signaling on NNV infection. IGF1R can activate GTPases,

such as Cdc42 and Rac1, leading to rapid actin reorganization and membrane ruffling, key processes in initiating macropinocytosis (*Dam et al., 2017*; *Manara et al., 2016*). Increasing evidence has demonstrated that Cdc42 and Rac1 are essential for the entry of various viruses (*Lu et al., 2023*; *Swaine and Dittmar, 2015*; *Zhang et al., 2021*). We found that both MmRac1 and MmCdc42 interact with the intracellular domain of MmIGF1R, and inhibition of MmRac1 and MmCdc42 significantly impaired RGNNV entry. Conversely, overexpression of MmRac1 or MmCdc42 enhanced NNV internalization. Actin filaments play a central role in driving the membrane dynamics necessary for the formation and internalization of macropinosomes during macropinocytosis. We found that there was a notable reorganization of F-actin filaments, accompanied by the presence of actin-driven membrane protrusions that co-localized with virions on the surface of RGNNV-infected cells. All these results indicated that cytosis MmIGF1R regulates Rac1/Cdc42 signaling to promote macropinocytosis during NNV infection. While previous studies have suggested that NNV enters host cells via macropinocytosis, our study was the first to provide a detailed mechanistic understanding of this process. The identification of MYL3 as a novel receptor for RGNNV entry, coupled with its role in activating the IGF1R-Rac1/Cdc42 axis to drive macropinocytosis, opens new avenues for therapeutic intervention in aquaculture. MYL3, as a crucial receptor for viral entry, could be inhibited through the development of small molecules, peptides, or monoclonal antibodies that block its interaction with the virus. Additionally, IGF1R inhibitors could prevent macropinocytosis and viral entry by disrupting downstream signaling pathways. Screening libraries of small molecules and natural products for potential inhibitors and testing them in aquaculture models would help evaluate their efficacy. Combining MYL3 and IGF1R inhibitors might offer a synergistic approach to prevent viral infection. These strategies could significantly reduce viral loads and improve fish survival rates, addressing the ongoing viral challenges in aquaculture.

Overall, MmMYL3 was identified as a novel functional receptor for RGNNV. Moreover, MmMYL3 was involved in and facilitated RGNNV internalization through the macropinocytosis pathway via the IGF1R-Rac1/Cdc42 axis. Our findings shed new light on the molecular mechanism of RGNNV entry and provide a new perspective for the development of antiviral therapies.

While our study provides strong evidence that MmMYL3 serves as a receptor for RGNNV in vitro, one limitation of our research is the lack of in vivo experiments using MmMYL3 knockout marine medaka. Conducting these knockout studies would further validate the role of MmMYL3 in NNV infection in a whole organism context. Unlike well-established model organisms like zebrafish or mice, gene editing in marine medaka is fraught with difficulties, including technical complexity, precise embryo microinjection, and the potential for off-target effects with CRISPR/Cas9 technology. The process is time-consuming, requiring multiple breeding generations and thorough molecular screening to establish homozygous knockout lines. Future research aimed at developing these knockout models will greatly benefit from advances in gene editing technologies and standardized protocols tailored to marine medaka.

## Materials and methods

### Cell culture

hMMES1 cell line, derived from marine medaka embryo blastocysts, was demonstrated to be susceptible to RGNNV (*Zhang et al., 2024*). hMMES1 cells were cultivated in ESM4 medium at 28°C as previously described (*Zhang et al., 2020*). HEK293T cells were cultured in DMEM supplemented with 10% heat-inactivated fetal bovine serum (Yeasen, China) under standard conditions of 37°C with 5% $CO_2$.

### Virus generation

The RGNNV strain, originally isolated from afflicted sea perch larvae and juveniles in Guangdong Province, China, was propagated in LJB cells (*Jia et al., 2015*; *Le et al., 2017*) and preserved as viral stocks at −80°C for use.

### Cell transfection and NNV infection

hMMES1 in six-well plates ($1\times10^6$ cells/well) were transfected with different plasmids using Lipofectamine 8000 (Beyotime) following the manufacturer's instructions. At 24 hr post-transfection, the cells were infected with RGNNV at a multiplicity of infection (MOI) of 1 for the indicated hours and examined by qRT-PCR.

To detect the viral titers, hMMES1 cells transfected with pCMV-Flag-MmMYL3 or MmMYL3-siRNA were infected with RGNNV (MOI=1) at 28°C for 4 hr. The culture medium was removed and incubated with fresh medium for 48 hr, then the supernatant was collected for viral titer assay as described previously (*Zhang et al., 2022b*).

## RNA interference

siRNAs targeting MmMYL3 (siMmMYL3) were synthesized by the RiboBio company (Guangzhou, China), including siRNA-1 (5′-GGCTCAACTTTGACGCCTT-3′), siRNA-2 (5′-CTGAGCTGCGAATTGT GCT-3′), siRNA-3 (5′-GCTGTAGCTTGCCTACAGA-3′), and control siRNA (NC) (5′-UUCUCCGAACGU GUCACGUTT-3′). siRNAs targeting MmIGF1R (siMmIGF1R) were siRNA-4 (5′-GCACAACTTGACAATC CGA-3′), siRNA-5 (5′-CGTCACATCAAGTTATACA-3′), siRNA-6 (5′-GCAATCTGACGTACTACCT-3′). Transfection of siRNAs was conducted using a mixture of three different siRNAs at concentrations of 50 or 100 nM as described previously (*Zhang et al., 2022b*).

## qRT-PCR

Total RNA of cultured cells was extracted with TRIzol reagent (Invitrogen, CA, USA) according to the manufacturer's instructions and was further reverse-transcribed into cDNA through the GoScript Reverse Transcription Mix (Promega, Madison, WI, USA). A LightCycler 480 II (Roche Applied Science, Germany) and qPCR SYBR Green Master Mix (Yeasen) were used for qRT-PCR analysis by using gene-specific primers. mRNA relative expression levels were evaluated from triplicate experiments and normalized to marine medaka β-actin. The relative fold induction of genes was calculated using the threshold cycle ($2^{-\Delta\Delta CT}$) method and presented as mean ± standard deviation (SD).

## Western blot and Co-IP

The cells were lysed with lysate buffer (Beyotime) and boiled for 10 min with 1% SDS for SDS-PAGE separation. The proteins were transferred onto polyvinylidene difluoride membranes (Millipore, USA) and then blocked with 5% nonfat dried milk for 1 hr at room temperature (RT), followed by incubation with primary antibodies at 4°C overnight, including anti-Flag (1:4000), anti-Myc (1:4000), anti-GFP (1:4000), anti-MYL3 (1:1000), and anti-actin (1:4000) antibodies. Subsequently, membranes were further probed with goat anti-rabbit or anti-mouse IgG (H+L) highly cross-adsorbed secondary antibodies (1:1000) for 1 hr at RT and analyzed using enhanced chemiluminescence immunoblotting detection reagents (Millipore, USA) on a chemiluminescence instrument (Sage Creation, China).

For the Co-IP assay, cell extracts were incubated with anti-Flag/Myc/GFP magnetic beads at 4°C overnight. After incubation, the beads were washed five times with lysis buffer and eluted with 1% SDS buffer for boiling. The eluted samples were subjected to western blot analysis.

## Protein purification and pull-down assays

The pET-GST-MmMYL3 and empty pET-GST were transformed into *Escherichia coli* BL21 (DE3), separately. Bacteria were grown to an $OD_{600}$ of 0.8–1.0 and then induced with 0.5 mM IPTG for 12 hr at 20°C in a shaking incubator. MYL3-GST protein and GST protein were purified using the GST spin purification kit (Beyotime) according to the manufacturer's instruction. Briefly, the bacterial solution was collected and washed three times with PBS. Lysozyme was added to a final concentration of 1 mg/mL and placed on ice for 30 min, then centrifuged at 4°C for 10 min. The supernatant was purified using the BeyoGold GST-tag Purification Resin as described previously (*Lv et al., 2022*). Finally, the protein was eluted from the column. The purified protein was stained with Coomassie brilliant blue.

GST-Tag magnetic beads were first mixed with the GST-fused MmMYL3s for 4 hr at RT. Then, the beads were incubated with protein lysates from HEK293T cells transfected with pCMV-Flag-CP or p-EGFP-MmIGF1R at 4°C overnight and finally analyzed by western blotting. For the pull-down assay between MmMYL3 and RGNNV, the beads incubated with GST-fused proteins were mixed with RGNNV viral stock (500 µL) at 4°C for 4 hr. Then, the beads were subjected to SDS-PAGE.

## SPR analysis

Biacore T200 (GE, USA) was used to analyze the interaction between MmMYL3 and the CP. Briefly, purified CP was first diluted (20 µg/mL) with 200 µL of sodium acetate buffer (10 mM, pH 4.0) and then flowed across the activated surface of a CM5 sensor chip (BR100012, GE, USA) at a flow rate of 10 µL/

min, reaching a resonance unit (RU) of ~2500. The remaining activated sites on the chip were blocked with 75 µL of ethanolamine (1 M, pH 8.5). Serial concentrations of MmMYL3 protein (5, 10, 20, 40, 80 µM) in PBS buffer (0.01 M, pH 7.4) were applied to analyze their interactions with immobilized CP at a flow rate of 10 µL/min. The $K_D$ for binding and Ka and Kd rate constants were determined using the BIA evaluation program (GE, USA).

## Proteinase K protection assay

The proteinase K protection assay was performed as described previously with some modification (*Zhang et al., 2020*). Briefly, hMMES1 and HEK293T cells were seeded into six-well culture plates and transfected with pCMV-Flag-MmMYL3 plasmids for 24 hr. After being washed with PBS three times, one group of cells was treated with 10 µg/mL of proteinase K for 30 min in an ice water bath, and another group was added with the same volume of PBS. The reaction was stopped by the addition of PMSF. Cells were then lysed with lysis buffer on ice for 30 min. The whole cell lysates were resuspended in SDS loading buffer and analyzed by western blot using anti-Flag or anti-Actin abs.

## Flow cytometry

Cells were seeded on six-well plates overnight and scraped off the plate and washed with PBS. For detection of surface expression of MmMYL3, cells were suspended with 200 µL PBS and incubated with anti-Alexa Fluor 488-Myc antibody at 2 µg/mL on ice for 2 hr. Cells were then washed with PBS and analyzed by flow cytometry.

## Transmission electron microscopy

Cell samples and ultrathin sections for TEM were prepared as described previously (*Dong et al., 2014*). Cells were collected and fixed at 4 °C for 24 hr with 2.5% glutaraldehyde in 0.1 M PBS (pH) and 2.0% osmium tetroxide in 0.1 M PBS; in turn, ultrathin sections were observed under a Philips CM10 transmission electron microscope.

## IF assays

To examine the localization of MmMYL3 proteins on cells, IF assays were performed as described previously with some modification (*Jia et al., 2013*). Briefly, hMMES1 and HEK293T cells were seeded into 24-well culture plates on glass coverslips and were separately transfected with pCMV-Myc and pCMV-Myc-MmMYL3 plasmids, respectively. After transfection for 24 hr, cells were washed with PBS, fixed with 4% paraformaldehyde for 10 min at RT. One group was treated with 0.2% Triton X-100 for membrane permeabilization, and the other was not. After being washed three times with PBS, cells were blocked with PBS containing 5% bovine serum albumin at RT for 1 hr and then reacted with anti-Myc abs (1:400) at 4°C overnight. Anti-Actin abs were used as a negative control. After three washes with PBS, cells were incubated with Alexa Fluor 555 goat anti-mouse IgG (Invitrogen) at a dilution of 1:400 for 1 hr at RT. Cells were then washed with PBS and stained the cell nuclei with DAPI for 10 min. The coverslips were washed with PBS and observed under an SP8 Leica laser confocal microscopy imaging system (SP8, Leica, Germany).

For assessment of the co-localization, HEK293T cells plated on coverslips in 24-well plates were transfected with different plasmids. After transfection for 24 hr, cells were washed with PBS, then fixed and permeabilized as described above. The cells were incubated with primary antibodies at 4°C overnight, including anti-Flag (1:400) and anti-Myc (1:400) antibodies, followed by incubation with Alexa Fluor 555- or 488-conjugated secondary antibodies against mouse or rabbit IgG (1:400). Finally, cells were observed under a confocal microscope.

For cell membrane staining, hMMES1 cells were labeled with PKH 26 (red fluorescent, MCE). PKH 26 was dissolved in DMSO to obtain a 1 mM stock solution of PKH 26. The stock solution was diluted with PBS at 1:100 to obtain the 10 µM working solution. hMMES1 cells were incubated with the working solution at 28°C for 30 min. After the incubation, the cells were washed using PBS to remove unlabeled cells and detected as described above.

For RGNNV entry detection, HEK293T cells were transfected with pCMV-Flag-MmMYL3 for 24 hr, and then infected with RGNNV (MOI=5). After infection for 24 hr, the cells were detected with IF assays, and cells were incubated with rabbit anti-CP abs and detected as described above.

For RGNNV entry detection, HEK293T cells were transfected with pCMV-Flag-MmMYL3 for 24 hr, and then infected with RGNNV (MOI=5). After infection for 24 hr, the cells were detected with IF assays, and cells were incubated with rabbit anti-CP abs and detected as described above.

## Dextran uptake assay for macropinocytosis

Following a published procedure (*Zhang et al., 2022a*), hMMES1 cells on coverslips were treated with the indicated chemical compound. After treatment, the cells were serum-starved for 16 hr. The cells were infected with RGNNV (MOI=1) in media containing Alexa Fluor 647-conjugated dextran (10,000 MW), incubated for 4 hr at 28°C, and washed and fixed for 16 hr with 4% PFA at 4°C. For co-localization experiments, the cells were also stained for MmMYL3 and CP using specific antibodies as described above. Samples were viewed and evaluated by confocal microscopy.

## Blocking assays

MYL3-GST protein was affinity-purified as described above. RGNNV ($10^3$ $TCID_{50}$) were incubated with different concentrations (100 or 500 ng) of recombinant MYL3-GST protein for 4 hr at 4°C. Then, hMMES1 cells were incubated with virus and protein mixtures for 4 hr at 4°C. Similarly, cells were treated with $10^3$ $TCID_{50}$ of RGNNV preincubated with BSA (500 ng) as a control. Then, cells were harvested, and CP and RDRP mRNA were measured as described above.

hMMES1 cells were pre-seeded in 12-well plates overnight. Due to the unavailability of anti-MmMYL3 abs and the high homogeneity of MYL3 at the N-terminus domain between human and fish, cells were incubated with anti-human MYL3 abs (1:50) (Abcam) for 4 hr at 28°C. After being washed with fresh media, cells were infected with RGNNV (MOI=1) at 4°C for 2 and 4 hr, respectively. As a control, hMMES1 cells were in parallel pretreated with normal rabbit IgG. Cells were then washed three times with PBS to remove free virus particles and harvested for total RNA extraction and qRT-PCR detection of RGNNV CP and RDRP.

To evaluate the implication of arm (short peptide of CP, RTDAPVSKASTVTGFGRG, Sangon, Shanghai) on RGNNV entry, hMMES1 cells were pretreated with different doses (0, 5, 10, 20, 40 μM) of these samples for 4 hr. Then, the virus (MOI=1) was subsequently added to allow infection for 4 hr. The virus-protein mixture was removed, and cells were further cultured with fresh protein-containing medium for 24 hr. The qRT-PCR protocols were performed as described above.

## Inhibitor treatment assay

Cells were treated with various inhibitors to investigate the endocytosis pathway utilized by RGNNV for cell entry. hMMES1 cells or MmMYL3-overexpressing hMMES1 cells were treated with different inhibitors for 4 hr, then infected with RGNNV (MOI=1) for 4 hr at 28°C. Next, the cells were washed with PBS three times to remove any unbound viruses. DMSO was used as control. Total RNA of cells was extracted for qRT-PCR detection.

## RGNNV infection of marine medaka

Marine medaka were divided into two groups: the MmMYL3 group and the GST group (n = 30 per group). Correspondingly, RGNNV (100 $TCID_{50}$) was incubated with purified GST-tagged MmMYL3 recombinant protein or GST protein for 4 hr at 4°C, then the mixtures were injected into two groups of marine medaka separately. The negative control group of fishes was injected with the same volume of PBS. The survival rate of each group was recorded every day by counting the numbers of dead fish. The log-rank test method was used to analyze the differences between groups. Marine medaka were necropsied, and the eyes and brains were collected at day 5 post-infection for further measurement of viral loads and pathological analysis. **, $p<0.01$. The animal study was reviewed and approved by the Ethics Committee of Sun Yat-Sen University (Approval No. SYSU-IACUC-2023-000168).

## Quantification and statistical analysis

Data were collected from three independent experiments, analyzed with SPSS version 20.0, and presented as the means ± SDs. Student's t-test or one-way analysis of variance (ANOVA) was used for the statistical comparisons between two groups or multiple groups, respectively. p-Value<0.05 was considered to indicate a statistically significant difference; p-Value<0.01 was considered to indicate a highly significant difference.

## Acknowledgements

This work was supported by the National Natural Science Foundation of China (32173001; 32473189), Guangdong Province Special Support Plan Youth Top Talent Project (NIQN2024002), Natural Science Foundation of Guangxi Province (2021GXNSFDA075015), Scientific and Technological Planning Project of Guangzhou City (2023B03J1267), Natural Science Foundation of Guangdong Province (2023B1515120074; 2024A1515010880), and Open Project Program of State Key Laboratory of Biocontrol (2023SKLBC-KF03).

## Additional information

### Funding

| Funder | Grant reference number | Author |
| --- | --- | --- |
| National Natural Science Foundation of China | 32173001 | Kuntong Jia |
| National Natural Science Foundation of China | 32473189 | Kuntong Jia |
| Guangdong Province Special Support Plan Youth Top Talent Project | NIQN2024002 | Kuntong Jia |
| Natural Science Foundation of Guangxi Zhuang Autonomous Region | 2021GXNSFDA075015 | Kuntong Jia |
| Scientific and Technological Planning Project of Guangzhou City | 2023B03J1267 | Kuntong Jia |
| Natural Science Foundation of Guangdong Province | 2023B1515120074 | Meisheng Yi |
| Natural Science Foundation of Guangdong Province | 2024A1515010880 | Wanwan Zhang |
| Open Project Program of State Key Laboratory Biocontrol | 2023SKLBC-KF03 | Kuntong Jia |

The funders had no role in study design, data collection and interpretation, or the decision to submit the work for publication.

### Author contributions

Lan Yao, Data curation, Formal analysis, Validation, Investigation, Visualization, Methodology, Writing – original draft, Writing – review and editing; Wanwan Zhang, Formal analysis, Funding acquisition, Validation, Investigation, Visualization, Methodology, Writing – original draft, Writing – review and editing; Xiaogang Yang, Formal analysis, Investigation, Methodology; Meisheng Yi, Conceptualization, Supervision, Funding acquisition, Project administration, Writing – review and editing; Kuntong Jia, Conceptualization, Supervision, Funding acquisition, Investigation, Writing – original draft, Project administration, Writing – review and editing

### Author ORCIDs

Lan Yao https://orcid.org/0009-0003-6725-4195
Kuntong Jia https://orcid.org/0000-0001-7674-9454

### Ethics

All experiments were conducted in compliance with ethical regulations. The fish experiments were carried out under the guidance of the European Union Guidelines for the Handling of Laboratory

Animals (2010/63/EU) and approved by the Ethics Committee for Animal Experiments of Sun Yat-sen University (No. 20240510002).

Reviewer #1 (Public review): https://doi.org/10.7554/eLife.104772.4.sa1
Reviewer #2 (Public review): https://doi.org/10.7554/eLife.104772.4.sa2
Reviewer #3 (Public review): https://doi.org/10.7554/eLife.104772.4.sa3
Author response https://doi.org/10.7554/eLife.104772.4.sa4

---

# Additional files

## Supplementary files

MDAR checklist

## Data availability

All data generated or analysed during this study are included in the manuscript, figures, figure supplements and source data files. source data files have been provided for Figures 1–7.

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
