## [Editor Report · eLife Assessment]

The findings in this manuscript are **fundamental** because they identify an entry receptor MYL3 that belongs to the myosin family as a possible target that could inhibit a virus that has a high impact on aquaculture. The evidence is **convincing** as it contains strong in vitro and in vivo data that support their conclusions; however, studies on the presence of MYL3 in NNV target tissues will further strengthen their claims

---

## [Referee Report · Reviewer #1 (Public review)]

Summary:

In this manuscript, the authors discovered MYL3 of marine medaka (Oryzias melastigma) as a novel NNV entry receptor, elucidating its facilitation of RGNNV entry into host cells through macropinocytosis, mediated by the IGF1R-Rac1/Cdc42 pathway.

Strengths:

In this manuscript, the authors have performed in vitro and in vivo experiments to prove that MnMYL3 may serve as a receptor for NNV via macropinocytosis pathway. These experiments with different methods include Co-IP, RNAi, pulldown, SPR, flow cytometry, immunofluorescence assays and so on. In general, the results are clearly presented in the manuscript.

Comments on revisions:

The authors have addressed all my comments.

---

## [Referee Report · Reviewer #2 (Public review)]

Summary:

The manuscript offers an important contribution to the field of virology, especially concerning NNV entry mechanisms. The major strength of the study lies in the identification of MmMYL3 as a functional receptor for RGNNV and its role in macropinocytosis, mediated by the IGF1R-Rac1/Cdc42 signaling axis. This represents a significant advance in understanding NNV entry mechanisms beyond previously known receptors such as HSP90ab1 and HSC70. The data, supported by comprehensive in vitro and in vivo experiments, strongly justify the authors' claims about MYL3's role in NNV infection in marine medaka.

Strengths:

(1) The identification of MmMYL3 as a functional receptor for RGNNV is a significant contribution to the field. The study fills a crucial gap in understanding the molecular mechanisms governing NNV entry into host cells.

(2) The work highlights the involvement of IGF1R in macropinocytosis-mediated NNV entry and downstream Rac1/Cdc42 activation, thus providing a thorough mechanistic understanding of NNV internalization process. This could pave the way for further exploration of antiviral targets.

Comments on revisions:

The authors have addressed the concerns from reviewers. This manuscript can be published in the current form.

---

## [Referee Report · Reviewer #3 (Public review)]

Summary:

The manuscript presents a detailed study on the role of MmMYL3 in the viral entry of NNV, focusing on its function as a receptor that mediates viral internalization through the macropinocytosis pathway. The use of both in vitro assays (e.g., Co-IP, SPR, and GST pull-down) and in vivo experiments (such as infection assays in marine medaka) adds robustness to the evidence for MmMYL3 as a novel receptor for RGNNV. The findings have important implications for understanding NNV infection mechanisms, which could pave the way for new antiviral strategies in aquaculture.

Strengths:

The authors show that MmMYL3 directly binds the viral capsid protein, facilitates NNV entry via the IGF1R-Rac1/Cdc42 pathway, and can render otherwise resistant cells susceptible to infection. This multifaceted approach effectively demonstrates the central role of MmMYL3 in NNV entry.

---

## [Author Response]

The following is the authors’ response to the previous reviews

**Reviewer #1:**
Specificity of MYL3 Selection:My previous question focused on why MYL3 was prioritized over other myosin family members. While the response broadly implicates myosins in viral entry, it does not justify why MYL3 was specifically chosen. For clarity, the "Introduction sections" should explicitly state the unique features of MYL3 (e.g., domain structure, binding affinity, or prior evidence linking it to NNV) that distinguish it from other myosins.

Thank you for your valuable comment regarding the specificity of MYL3 selection. In response, we have revised the "Introduction" section to explicitly clarify the rationale for prioritizing MYL3 over other myosin family members. Specifically, we have now included prior evidence linking MYL3 to NNV infection, citing our studies that identified MYL3 as a potential host factor interacting with NNV CP protein. In our previous study, sixteen CP-interacting proteins were identified by Co-IP assays followed by MS, including HSP90ab1, Centrosomal protein 170B, MYL3 and so on. In addition to our findings, previous study by other researchers has also reported that *Epinephelus coioides* MYL3 can bind to NNV (page 3, lines 79–81). These revisions provide a clearer justification for the selection of MYL3 and distinguish it from other myosin proteins. The added content can be found in the revised manuscript on page 3, lines 81–84.